# Simulation of Leather Visco-Elastic Behavior Based on Collagen Fiber-Bundle Properties and a Meso-Structure Network Model

**DOI:** 10.3390/ma14081894

**Published:** 2021-04-10

**Authors:** Sascha Dietrich, Olga Lykhachova, Xiaoyin Cheng, Michael Godehardt, Markus Kronenberger, Michael Meyer, David Neusius, Julia Orlik, Katja Schladitz, Haiko Schulz, Konrad Steiner, Diana Voigt

**Affiliations:** 1FILK Freiberg Institute gGmbH, Meißner Ring 1–5, 09599 Freiberg, Germany; sascha.dietrich@filkfreiberg.de (S.D.); michael.meyer@filkfreiberg.de (M.M.); haiko.schulz@filkfreiberg.de (H.S.); diana.voigt@filkfreiberg.de (D.V.); 2Department of Flow and Material Simulation, Fraunhofer Institute of Industrial Mathematics (ITWM), Fraunhofer-Platz 1, 67663 Kaiserslautern, Germany; olga.lykhachova@itwm.fraunhofer.de (O.L.); david.neusius@itwm.fraunhofer.de (D.N.); julia.orlik@itwm.fraunhofer.de (J.O.); konrad.steiner@itwm.fraunhofer.de (K.S.); 3Department of Image Processing, Fraunhofer Institute of Industrial Mathematics (ITWM), Fraunhofer-Platz 1, 67663 Kaiserslautern, Germany; xiaoyin.cheng@itwm.fraunhofer.de (X.C.); michael.godehardt@itwm.fraunhofer.de (M.G.); markus.kronenberger@itwm.fraunhofer.de (M.K.)

**Keywords:** multi-scale simulation, hierarchical leather structure, collagen fiber-bundles, stress-strain experiments, micro computed tomography, microscopy imaging techniques, FEM, Voronoi tessellation

## Abstract

Simulation-based prediction of mechanical properties is highly desirable for optimal choice and treatment of leather. Nowadays, this is state-of-the-art for many man-made materials. For the natural material leather, this task is however much more demanding due to the leather’s high variability and its extremely intricate structure. Here, essential geometric features of the leather’s meso-scale are derived from 3D images obtained by micro-computed tomography and subsumed in a parameterizable structural model. That is, the fiber-bundle structure is modeled. The structure model is combined with bundle properties derived from tensile tests. Then the effective leather visco-elastic properties are simulated numerically in the finite element representation of the bundle structure model with sliding contacts between bundles. The simulation results are validated experimentally for two animal types, several tanning procedures, and varying sample positions within the hide. Finally, a complete workflow for assessing leather quality by multi-scale simulation of elastic and visco-elastic properties is established and validated.

## 1. Introduction

Leather is a renewable material that is primarily used in manufacturing of high-priced consumer goods like shoes, clothing, upholstered furniture, or automotive interiors. Due to the naturally grown raw material from animals’ skins, each up-cycled tanned hide for leather production is unique. Being proclaimed as a marketing advantage on the one hand, this high variability represents, on the other hand, a considerable challenge for production, processing, and final application of leather.

It is well-known that all biological tissues and materials are visco-elastic, see for example, [1,2]. Thus, leather does not only respond immediately by deformation to an applied load, but also creeps. The deformation increases further if the loading is kept for a while. The difference to a simple plastic deformation lies in the relaxation behaviour. This means that, after removing the load, the deformation starts to decrease, and completely disappears after a certain time-period. The visco-elastic or relaxation, creep properties of leather play a huge role in, for example, formation of wrinkles on leather seat covers under cyclic loading, see for example, [3].

The collagen-based hierarchical 3D structure of the animal hide is essentially retained in manufacturing. Affecting the properties of the finished leather, the native structure causes a material-inherent large variance and pronounced anisotropy of the physical properties [4] due to (micro)structural variations [5,6] depending on species, race, gender, age, husbandry conditions, as well as individual body parts and tanning processes [7]. This pronounced variability together with the discontinuous production and processing increase the need for general characteristics for leather qualification and structure-related information for specific leather applications.

Structural composition, as well as micromechanical properties of leather materials, are mainly accessible by extensive material-destroying physical measurements and a variety of microscopic imaging techniques. Recently, non-destructive testing methods, like ultrasound imaging, small-angle X-ray scattering, and computed tomography (CT) have been applied to capture structural features of the collagen fiber-bundles [8,9,10]. Truly, 3D quantitative geometric analysis of leather samples based on CT image data has been reported in [10,11]. The 3D geometries are compared in [10] based on separation into solid (collagen) and pore structure. However, a robust, generally applicable method for segmentation of individual fiber-bundles and their connectivity is still missing in spite of dedicated algorithms developed [11,12]. Here, a parameterizable model for this meso-scale bundle structure is introduced. This network model formed by interwoven 2D tessellations captures essential qualitative features observed in the 3D images.

Leather properties are determined significantly by the fiber-bundle structure and their elastic and relaxation properties [1,13]. The meso-structure network observed in the 3D images is small compared to the true length of the collagen fiber-bundles forming it. Thus, direct numerical computation of elasticity on the resolved meso-structure is not feasible. A multi-scale asymptotic approach separating the scales and providing effective macroscopic material behavior can be used to reduce the dimension of the problem, [14,15,16,17,18]. The effective moduli can be obtained from solutions of auxiliary problems with frictional contact on the representative volume element [14,15,19,20,21]. The thickness of fibers is small compared to their length. Thus, the dimension of the auxiliary problems in the representative cell can be reduced further by an asymptotic approach with respect to the fiber thickness [19,22,23,24]. Finally, it is numerically solved by the finite element method with frictional contact [20,21], and extended to visco-elastic relaxation [25,26].

The approach sketched above relies on a geometric model for the collagen fiber-bundle network reproducing contacts or branching, in particular the winding-contact-branching. This essential geometric information is deduced from 3D image data obtained by micro-computed tomography of leather snippets of the size of a few millimeters. 3D quantitative analysis of the network structure based on these images emphasizes the high variability with respect to animal, position within the hide, and tanning. Segmentation and successive analysis of representative numbers of fiber-bundle segments and contact elements failed, as all attempts reported on previously [10,11,27,28] lacked sufficient robustness and generalizability. Nevertheless, qualitative features of network and branching could be read off the 3D images. These are incorporated in our novel network model consisting of interwoven dilated edge systems of 2D Poisson-Voronoi tessellations [29].

Together with experimentally determined (visco-)elastic and relaxation properties of fiber bundles, the network model yields a meso-structural simulation tool for comparing several species, races, as well as individual body parts and tanning processes.

## 2. Materials and Methods

### 2.1. Tensile Experiments on Leathers and Fiber-Bundles

Leather samples and fiber bundles from back, flank, and axilla of metal-free vegetable-tanned, non-dyed bovine hides from the Simmental breed (leathers L1 and L2), zebu breed (leather L3), and synthetically tanned bovine hide (leather L4) were investigated (FILK tanning laboratory, Freiberg, Germany). All leather samples were reduced to their reticular layers by splitting off the grain and papillary layers. The sample conditioning and preparation followed ISO 2418, ISO 2419, and ISO 2420.

Light and electron microscopic imaging of complete leather structure areas, as well as selected substructures and structure elements like collagen fibers and collagen fiber-bundles confirmed the hierarchical leather structure, as well as gradients in density (Figure 1 and Figure 2). Weave angle, density, orientation, fiber-bundle thickness, and fiber-bundle shape can be deduced for qualitative, rough structure description. The intrinsic fiber-bundles appear wavy and crimped. However, actual parameterized values are not accessible by these techniques (see Figure 3). Thus, samples for µCT measurements are picked from each leather L1–L4 and all sampling positions (back, flank, axilla) for further analyses.

Macroscopic bulk densities and thicknesses are determined according to ISO 2420 and ISO 2589, respectively. For leathers L1–L3, identical technological processing resulted in similar raw densities (0.51 to 0.71 g/${\mathrm{cm}}^{3}$) and thicknesses (1.35 to 1.86 mm) (see Table 1). Results for leather L4 are in the same order of magnitude. Tensile strength σ [MPa] and elongation ϵ [%] are measured at maximum force $F_{max}$ [N]. Young’s modulus *E* [MPa] are recalculated as a slope of the linear part of the stress-strain curves.

#### 2.1.1. Tests on Fiber-Bundles

The micro-mechanical properties of collagenous fibers and fiber-bundles have to be measured as they determine the macroscopic leather material behavior to a high extent. To this end, collagen fiber-bundles are gently manually separated from the back, flank, and axilla of leathers L1–L4. They are imaged by light microscopy to determine their cross-sectional shapes and diameters (see Figure 3). The thicknesses (diameters) were determined as averaged values (see Table 2) of 3–5 measurements along the length (Figure 3). Fiber-bundles are found to differ in length and fibrousness (e.g., compact or loose), but not significantly in diameter (see Table 2). The strength and elongation properties are given in the same designation as in Table 1.

Fiber-bundles are subject to two types of load scenarios: (1) Classical uniaxial tensile test up to fiber-bundle breakage, and (2) incremental loading up to a pre-defined elongation followed by recording the force drop. Relaxation results for fiber-bundles and leathers are discussed in Section 3.2 as well.

Individual micromechanical stress-strain experiments using a test stage of Kammrath & Weiss GmbH yield force and elongation parameters on fibre-bundle scale. All the experiments were conducted on samples that were 2 mm in length, clamped in the testing machine from both sides. Tension was applied by increasing displacements. Videos help to observe the tensile load-induced deformation process on a microstructure level, including complete fiber-bundle reorientation (Figure 4a,b), fiber elongation (Figure 4b,e), fiber slippage (Figure 4c,d), and final fiber-bundle breakage (Figure 4f). Figure 4 shows that the fibers forming the bundle break consecutively, when the fibers’ stress-strain limit is exceeded.

Typical force-elongation diagrams for loaded fiber-bundles are presented in Figure 5 for L1–L4 on samples from the back and flank, respectively. They feature the well-known J-shape [1]. The slope’s characteristics depend on the sample position within the hide, but not on animal or tanning type. Consequently, elastic moduli for the fiber-bundles are similar, although the strains are higher for L4 17–20% compared to 5–6% for L1–L3 (see Table 2).

#### 2.1.2. Tests on Leather Samples

Dumbbell-shaped leather specimen from the back, flank, and axilla are tested according to ISO 3376. Six specimens are chosen in each of the three sample positions within the hide: three specimens in longitudinal direction, parallel to the back line of the animal, and three specimens perpendicular to the back line (transverse direction). See Table 3 for averaged geometric parameters of the leathers. Here, strength properties are introduced as two values: (1) Averaged stresses $\sigma^{*}$ and $\sigma^{**}$ found for the strains of 20% and 30%, respectively for back and flank specimens; and (2) averaged maximum stresses (σ) corresponding to maximum elongation forces like in Table 1 and Table 2). Leathers L1–L3 exhibit almost similar tensile strengths and elongations, while the respective values for leather L4 are slightly higher. In general, the macroscopic tensile strength decreases in the order of back > flank > axilla. This observation can be linked directly to solid volume fraction (see details on the statistical analysis in Section 2.2 and corresponding Figure 8b).

Overall, 66 specimens were tested. Typical stress-strain diagrams (Figure 6 and in Section 3.1) for the studied leathers correspond qualitatively to the curves for the fiber-bundles (Figure 5). In general, transversal samples yield lower stresses and larger magnitudes of strain than longitudinal samples. In some cases, large differences can be observed for the leathers between the back, flank, and axilla.

Table 3 summarizes the data on tested leathers depending on animal and tanning type. Experimental results on leathers are similar to those for the corresponding fiber-bundles. Macroscopic elastic moduli of leathers are of the same order. Zebu samples (L3) have decreased axial stiffness compared to samples from Simmental breed (L1 and L2). The non-uniformity in elastic properties is less pronounced for the L4 samples which are, however, thinner than leathers L1–L3.

The zebu leather samples from L3 yield lower maximal stresses than Simmental breed samples (L1, L2), while L4 features the largest magnitudes. The strain level of L4 is higher (40–50%) than those of L1–L3 (20–30%, see Figure 6 and Table 3). The deformation curves of chrome-free leather L4 are smooth, while those of vegetable tanned leathers L1–L3 feature zigzags at higher forces.

### 2.2. Micro-Computed Tomography of the Meso-Scale Collagen Fiber-Bundle Networks

Micro-computed tomography for vegetable tanned leather has been successfully applied [10,11]. A custom-made CT device featuring a Feinfocus FXE 225.51 X-ray tube with maximum acceleration voltage 225 kV and maximum power 20 W and a Perkin Elmer flat bed detector XRD 1621 with 2.048 × 2.048 pixels was used. Imaging parameters were optimized to capture the low absorbing meso-structure optimally. The tube voltage was low (75 kV), and integration time high (1 s). Tomographic reconstructions were obtained from 1.200 projections, each averaged over 4. Thus, the overall integration time was 4 s. The pixel edge length of 3.3 μm allowed to capture the fiber-bundles, as well as parts of their substructure.

Geometric analysis of the samples based on binary images holding the solid collagen structure as foreground and the pore space as background (Figure 7) shows the expected high variation due to the animal, breed, and position within the hide (Figure 8). In order to capture the local variation, too, four virtual sub-samples from each 3D image were cut. The back samples clearly appear denser than those from the flank or axilla (Figure 8b). Leather 4 features a significantly higher surface-area per volume fraction $S_{V}$ and a much lower Euler number density $\chi_{V}$. The latter reflects branching point density, as the Euler number is the alternating sum of connected components, tunnels, and holes in the structure [30]. The holes are closed, and there is just one connected component. Hence, the Euler number density $\chi_{V}$ equals (1−#tunnels)/V. The denser the network gets, the more contacts between bundles appear per volume. This, in turn, increases the number of tunnels, and thus decreases the Euler number density. Leather 4 features, on the one hand, more connections, as measured by the Euler number density, and on the other hand, a larger surface area, particularly when compared to the zebu leather L3 (Figure 8). The latter could be attributed to loosening of the bundle structure (Figure 7) by mechanical processing. This explanation somewhat contradicts, however, the higher density of nodes. The meso-structure of leather is known to be anisotropic [4,31]. This was confirmed by the CT image analysis in [10], too. However, no significant deviations from isotropy of the structure are observed here. This rather surprising finding can be attributed to cutting the long connected networks of fiber-bundles during sample preparation for the CT measurement, leading to the loss of the pre-stress in the bundles and subsequent reorganization of the network.

The branching behavior is further investigated by the method from [32]: Branching point candidates were extracted from the 3D skeleton. Superfluous candidates were removed or united based on criteria read from the spherical granulometry map. The rationale behind this algorithm is that branching regions are slightly thicker than the surrounding structure. Vertices in overlapping largest spheres from the granulometry map were united. The valence of such a contracted branching point was then determined by counting the intersections of the skeleton with the spherical shell enclosing the branching region. Figure 9 visualizes this workflow using 2D slices from the 3D images.

Cheng’s method relies heavily on a nicely pruned skeleton, which in turn can be obtained from a smooth binary (black-and-white) image without tunnels or wholes within the component to be skeletonized. Thus, all bundle substructures captured in the μCT images has to be removed. Therefore, the original binary image, showing the bundle substructure, is smoothed again using a Gaussian filter with σ≈ bundle thickness. The blurred image is subsequently binarized again, such that the solid volume fraction does not change. That results in a virtual densification of the bundles that is compensated for by slightly reducing their thickness. As a side effect, branching regions are particularly thickened. The resulting smooth solid structure free of holes is skeletonized using the algorithm of [33].

Cheng’s algorithm contracts skeleton nodes. Consequently, the valence of these new nodes increases. Nevertheless, a clear predominance of Y nodes is observed.

### 2.3. Network Model

Structural modeling is needed to generate geometrically simple elements including proper connectivity. The network structure of the fiber-bundles features predominantly Y-shaped nodes, as described above. Moreover, the fiber-bundle network is forming essentially one connected component. These two observations motivate the choice of a structural model based on the edge system of a Voronoi tessellation. More precisely, the edge systems of planar Voronoi tessellations generated by macroscopically homogeneous, random Poisson point processes are rotated spatially, and interwoven. This allows for control by a very small set of parameters, captures the microscopic spatial variation, and naturally reflects the strong prevalence of Y nodes. The latter is due to the fact that vertices in planar macroscopically homogeneous random Poisson-Voronoi tessellations are almost surely of valence three [34].

The planar edge systems as shown in Figure 10 are dilated and rotated. Superpositions of sufficiently many 2D networks lead to a densely interwoven spatial fiber system. Examples are visualized in Figure 11.

The model is controlled by the mean number λ of points per area of the generating point process for the 2D tessellations, the mean number ν of 2D tessellations, the orientation distribution guarding the rotations, and the dilation thickness. Here, only λ and ν are varied, with the former mainly affecting the cell size and thus the branching point density, and the latter rather controlling the total edge length per volume and thus the volume fraction. The 2D networks are placed independently. Thus, the dilated networks can touch, as well as overlap. At these overlaps, a penalizing normal force and tangential frictional sliding is applied (see Section 3.2 for details). See Figure 10 and Figure 11 for realizations of 2D and 3D networks with varied λ and ν, respectively. For the leather, much denser networks are used. More precisely, λ=8 for the flanks of L2, L3, λ=12 for the respective back samples, and λ=25 for both the flank and back of L4. ν is adjusted to meet the solid volume fractions reported in Table 1. See an example for a realization of the model for the zebu (L3) back in Section 2.4.

### 2.4. Simulation Tool

The mechanical behavior of the leathers is modeled using our in-house software-tool FISFT which includes a 1D finite element method for large deformations and friction evolution, see [20]. Theoretical background on how to reduce the three-dimensional friction models to 1D contact conditions, is provided in [19,22]. The tool was extended in [21] to adhesion and complicated knots (Figure 12a, upper) and wound fibers in friction contact. Moreover, it allows to consider the influence of frictional slip evolution in local fiber-bundle knots on the macroscopic leather behavior (Figure 12a).

At low friction, bending or compression of the material does not influence the torsion at the contacts (see [25,26] for theoretical justification). Exceptions are knots (Figure 12a, right). If the contribution of torsion to the effective properties of the fibrous material is negligible, the relaxation of leather samples and their fiber-bundles are in agreement (see results of Section 3.2).

The meshes incorporate simple contact interactions between fiber-bundles (Figure 12b), as well as the Y branches found to be typical for leather (Figure 12c). The contacts are controlled by two parameters of normal and tangential forces. Proper choice of these parameters ensures a realistic fiber reorganization of leather meshes under tensional loading in horizontal direction. In Figure 13, the relative stress distribution locally acting within each element can be seen.

### 2.5. Simulation on a Meso-Structure of Non-Linear Elastic Fibers on the Scale of Several Millimeters

Based on the model described in Section 2.3, one can generate meshes with varying fiber volume fractions (see Figure 11). The real samples feature higher fiber volume fractions within 50 ÷ 70%, see Table 1. Meshes are generated iteratively. That is, 2D networks are added till the desired fiber volume. Penetrations or near penetrations are turned into contacts between the 2D networks. The stress-strain behavior of leather under tensile load is simulated on the small fiber-bundle meshes with an initial size of 5 × 5 × 5 ${\mathrm{mm}}^{3}$ (Figure 14). Experimental curves for the uniaxial tensile tests of fibers measured in Section 2.1 (Figure 5 and Table 2) are used as material input data. The results of the simulations are shown together with the experimental data for comparison in Figure 6 for Simmental breed and zebu, respectively. Clearly, the experimental curves end at higher stresses and strains compared to our numerical simulations. On the one hand, this is related to the high density of the meshes, which limits reorientation of the fibers during the loading process and thus the elongation of the leather meshes. The calculations are terminated deliberately when the maximum fiber-bundle strain is reached (according to Figure 5). In this range, the simulation agrees very well with the experimental data (Figure 6). On the other hand, differences in scale have to be kept in mind. The leather samples can elongate much longer (about 20–30%) than the fiber-bundles (5–10%) without much damage just by unfolding long continuous networks of folded fiber-bundles. Extraction of samples for the CT imaging cut the fiber-bundles, made them shorter compared to the tested fiber-bundles, and led to a reorganization of the fibers. Consequently, rather isotropic networks is obtained (see for details Section 3.1 and corresponding Figure 16).

## 3. Results and Discussion

### 3.1. Validation Based on Synthetically Tanned Leather and Discussion

The values for L4 in Table 1 and Table 2, and Figure 5 do not reveal a significant influence of the tanning on the properties of the fiber-bundles. Hence, the slope of the curves in Figure 5 is the same. Using the method described in Section 2.5, the properties of leather L4 are simulated with corresponding fiber volume rates of 66 and 68% for the back and flank. Results are compared with experimental data in Figure 15. Again, the simulation qualitatively repeats the experiments. FISFT curves for leather L4 end earlier and lie between the experimental results for the longitudinal (solid curves) and transverse samples (dotted curves).

Using the experimental and CT image data as inputs for numerical simulation of leathers L1–L3 and having verified the methodology for leather L4 of another tanning type, the main micro-structural features can now be deduced by influencing the macroscopic leather behavior. The leather behavior is decisively influenced by fiber volume fraction, fiber-bundle force-strain function, and number of contacts. The latter can be changed with the tanning process. The well-known macroscopic mechanical anisotropy cannot be reproduced based on the 3–5 mm sample range considered here. According to [31], arbitrary anisotropy can be achieved by tanning and introduction of boundary conditions (fixation and tension in certain directions). On the animal, the skin is fixed at six points, four paws, and head and rear parts. These six directions determine the fiber orientation [35] (in the pulling direction, see Figure 16a). Introducing new pulling directions, the anisotropy can possibly be altered. This is only possible with long contiguous and locally folded (interlaced) networks [36], which unfold or disentangle themselves during deformation and reorient themselves on the geometric scale of the whole leather piece. Only by this “freewheeling” (reorientation-unfolding) can the leather withstand much more deformation than the fiber-bundle.

These hypotheses are subject of further research. Nevertheless, already now, the leather behavior on the millimeter scale can be predicted very well, based on the fiber-bundle structure, the non-linear elastic properties of the fiber-bundles, and including the contact interaction of fiber-bundles in the numerical mechanical simulations. The simulation results agree very well with the real measurements on this scale.

This section is devoted to the static non-linear elastic behavior of the leather. The next section yields further insight into the relaxation behavior of the leather and its forecasting on the basis of the same structural and fiber-bundle properties.

### 3.2. Prediction of Visco-Elastic Properties of Leather Samples and Relaxation Tests

Visco-elastic properties are investigated experimentally in the static regime for the samples described in Section 2.1. A tensile load is increased incrementally up to a predefined elongation. Then, the specimen is held, and the force changes in time are recorded. This two-stage loading procedure is repeated till fracture. Fiber-bundles are torn to a predefined elongation with following relaxation only once. The relaxation behavior of fiber-bundles from L1 and L3 is shown in Figure 17. The fiber-bundles are stretched by a certain value compared to the initial state, and kept in this state for 180 s. The stresses depend on the position in the hide and decrease in the order of flank > axilla > back. In general, Simmental breed curves are smooth, with the lower level of initial stresses being comparable to the ones for zebu. The step-like stress drops, and increases of the zebu samples are due to slippage and/or fracture (see Figure 4b–d) of individual components in the fiber-bundle.

The results of relaxation tensile tests on leather samples with a square base area are shown in Figure 18. Here, each stretching step is held for several hours, and the relaxation is monitored. The relation of the observed relaxation behavior to the hierarchical relaxation properties of the fiber-bundles is investigated based on the assumption that the behavior curve for a single fiber-bundle multiplied by the purely elastic structure-geometric factor describes the macroscopic relaxation behavior of the leather [25,26]. This assumption is confirmed by the following comparison in Figure 19.

Here, the results for the samples of Simmental breed (Figure 19a) and zebu (Figure 19b) are presented for fixed initial elongation of about 20 and 5%, respectively. Blue curves refer to relaxation results on leather presented in Figure 18. Red curves correspond to fiber bundle curves similar to Figure 17 and multiplied by the ratio of simulated static Young’s moduli of the leather and its fiber-bundles. Hence, for the scale of a few millimeters, the relaxation behavior of leather is well-determined by relaxation of the fiber-bundles. This is due to the high density and low contact sliding [26].

### 3.3. Macro-Structural Simulation of Leather Damage

The previous sections yield the effective macroscopic properties of leather samples on the meso-scale using micro-structure and fiber-bundle properties. In order to reach the scale relevant for industrial applications, simulations are upscaled to larger samples entering the meso-simulation results from Section 2.5 as inputs. Here, tensile tests are simulated using ANSYS [37]. The following figures Figure 20 and Figure 21 illustrate a finite element simulation of leather fracture using a macroscopic orthotropic model with the previously calculated homogenized properties and local strengths. To simulate cutout growth under vertical tension, Birth/Death technology is used for those elements, whose stresses exceed the local strength limit. The full Newton-Raphson iterative procedure is used for geometrically non-linear static analysis. An alternating static tensile loading with verification of node stress variables is simulated in several loading steps. If the current equivalent von Mises stresses exceed the strength limit, elements that are connected to selected nodes are deactivated. Hereby, the corresponding “killed” elements are assigned a very small stiffness till the end of the simulation. As a first approximation, 3D 8-node structural finite elements SOLID185 with options of pure displacements and full integration is tested. The leather stripe is modeled as a cuboid of the dimensions 20.0×10.0×1.77
${\mathrm{mm}}^{3}$. The total element number is about 13,700, with 6 FE per thickness. The orthotropic properties are: Young’s moduli Ex/y/z=55/45/45 MPa, shear moduli G=20 MPa, and Poisson’s ration μ=0.3. The sample is loaded along the x-direction. The bottom is clamped, and uniform tension on the top is applied. The critical stress for element deactivation is bounded by 10 MPa. Sufficiently good qualitative results of the simulation are presented in Figure 20.

In the next step, leather is described using the shell theory. Finite element simulations are performed for about 7900 structural orthotropic 8-node shell elements SHELL281 with membrane stiffness. Geometric and mechanical parameters of the virtual leather samples are shown in Table 4. Two cases of clamped structures under tensile load are considered: An intact leather strip and a leather strip of the same dimensions, pre-damaged by a circular hole in the center. These simulations of leather fracture can be interpreted as progressive fiber separation and fractures leading to macro-damage in leather. The damage behavior of the two samples or the propagation behavior of the damage (Figure 21) differs significantly. The leather stripe without damage breaks near the boundary. In contrast, the initial damage in the other stripe is propagated.

Table 4 contains input data and results of the numerical calculation of two leathers (Simmental breed and zebu) exposed to static tensile loading. The same general geometry of samples with elastic properties, corresponding to similar leather types, is used.

## 4. Conclusions and Outlook

Based on high-resolution 3D geometric data from CT images, a stochastic structure model was suggested that captures the fiber-bundle branching on the meso-scale (millimeter scale) of leather very well. Realizations of this model enable mechanical simulation, accounting for frictional sliding and visco-elastic behavior of the leather on the basis of known (measured) behavior of its fiber-bundles. Simulations were based on advanced mathematical modelling, reducing the dimension of contact problems and numerical techniques, including complicated mechanical issues as frictional sliding at the complex contact junctions. All simulations were validated by mechanical tests and show very good agreement on the investigated scale.

The anisotropy of leather on several scales was discussed in detail in Section 2.2 and Section 3.1. The structural models introduced in this paper are well-suited for forecasting the leather behavior on the millimeter scale. This paper provided and justified the hypothesis that relaxation properties of leather on this scale are defined by just the bundle structure and the relaxation time of the bundles. In the last part, a technique for a multi-scale or just macroscopic simulative damage analysis was proposed, and the workflow within ANSYS, a commercial finite element tool for structural simulations, was described. Summarizing the results leads to a conjecture on the possible global network-structure of the complete leather and its local anisotropy due to frozen pre-strains in the bundles (Figure 16). Studying this hypothesis will help to understand the leather behavior completely and obtain a generalized model. In the long run, a simulation tool will allow leather companies to investigate leather properties easily.

## Figures and Tables

**Figure 1 materials-14-01894-f001:**
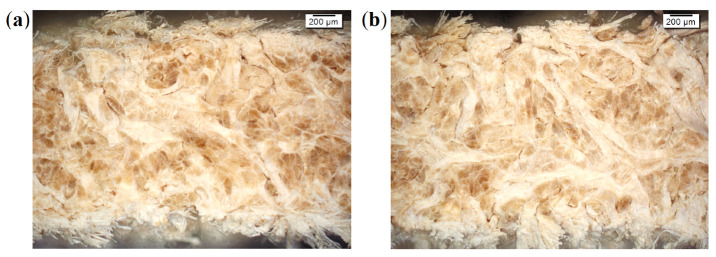
Light microscopic images of split leather samples from back parts: (**a**) Simmental breed leather L1 and (**b**) zebu breed leather L3.

**Figure 2 materials-14-01894-f002:**
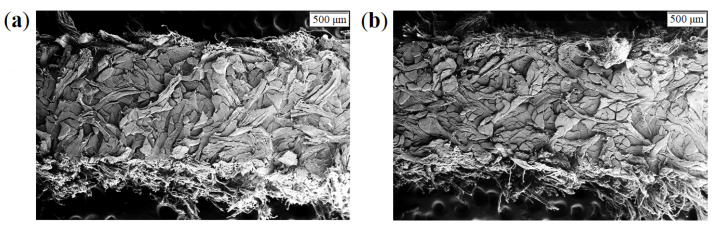
Electron microscopic images (secondary electron signal) of split leather samples from back parts: (**a**) Simmental breed leather L1 and (**b**) zebu breed leather L3.

**Figure 3 materials-14-01894-f003:**
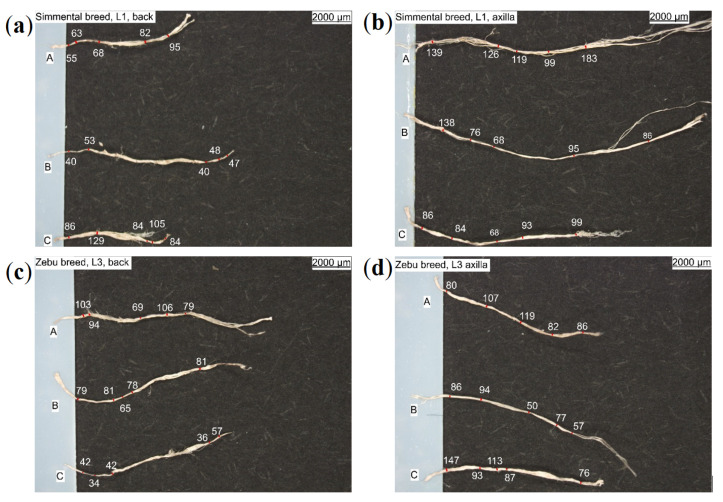
Light microscopic images including dimensioning of collagen fiber-bundles, extracted from (**a**,**c**) back and (**b**,**d**) axilla of (**a**,**b**) Simmental breed, L1 and (**c**,**d**) zebu, L3.

**Figure 4 materials-14-01894-f004:**
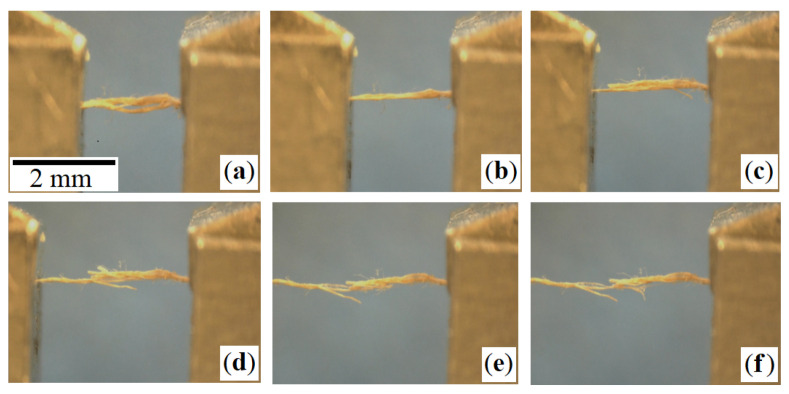
Illustrating uniaxial tensile test (load scenario 1) on a collagen fiber-bundle: (**a**) Initial state; (**b**) increased load causes fiber-bundle straightening and elongation; (**c**) slippage of fibers and beginning of fiber breakage; (**d**,**e**) progressing sequential fiber breakage; and (**f**) complete breakage of fiber-bundle.

**Figure 5 materials-14-01894-f005:**
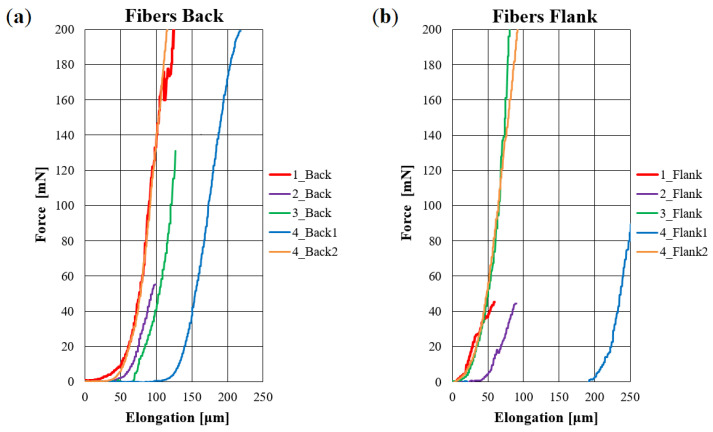
Comparative results of real tensile tests on fiber-bundles with elongations of 5-6% (L1–L3) and up to 17–20% (L4) corresponding to the strength limit: (**a**) Back and (**b**) flank.

**Figure 6 materials-14-01894-f006:**
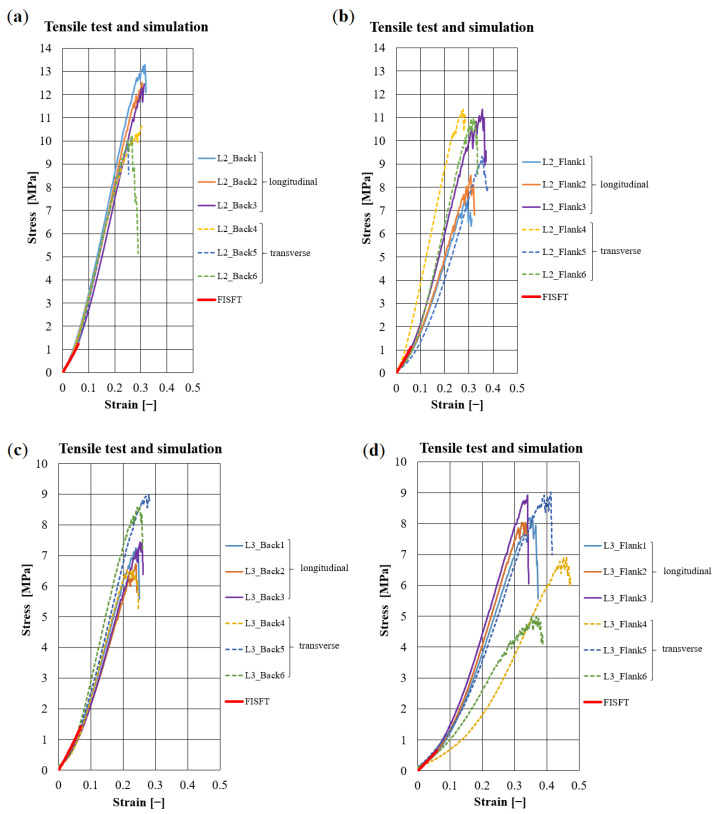
Comparison of experimental results with the stress simulation (FISFT) on meso-structural meshes representing leather samples of Simmental breed, L2: (**a**) back; (**b**) flank; and leather samples of zebu, L3: (**c**) back and (**d**) flank. Continuous red-colored thick lines represent results of numerical simulations; continuous thin lines are experiments for the longitudinal direction, and dotted thin lines for the transverse direction of the samples.

**Figure 7 materials-14-01894-f007:**
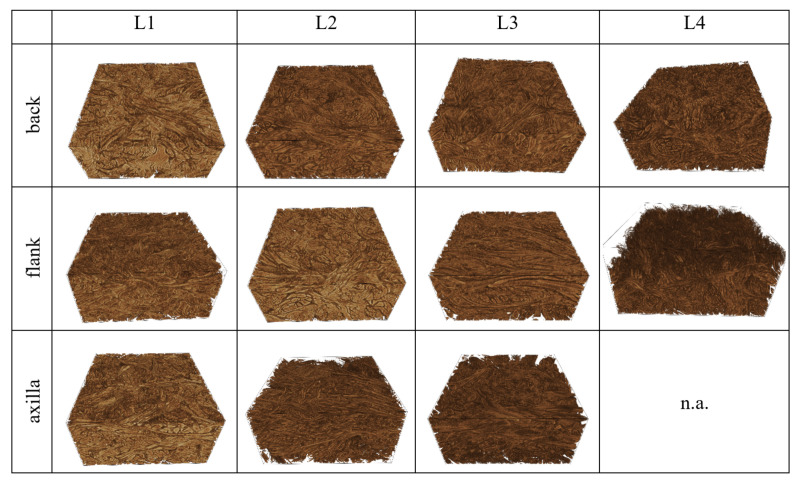
Renderings of (600 × 600 × 300 pixels corresponding to 2 × 2 × 1 ${\mathrm{mm}}^{3}$) subvolumes of the reconstructed images of all 11 investigated leather specimens.

**Figure 8 materials-14-01894-f008:**
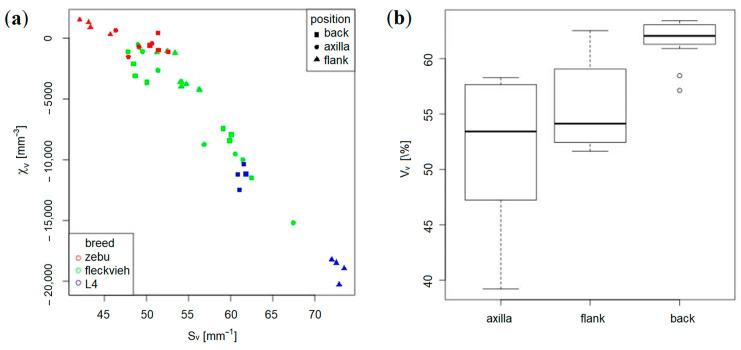
(**a**) Euler number density $\chi_{V}$ and specific surface area $S_{V}$ for the 11 × 4 investigated (sub-)samples and (**b**) solid volume fraction for each position within the hide.

**Figure 9 materials-14-01894-f009:**
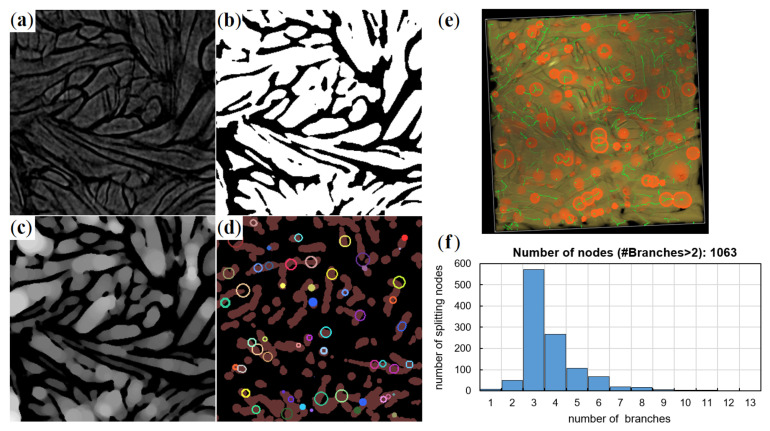
Workflow for branching point detection: (**a**–**d**) 2D slices from a 3D sub-volume of approximately 1 mm edge length of the reconstructed µCT image of L2, back. In clockwise order: (**a**) original; (**b**) binarized with solid fiber-bundles; (**c**) spherical granulometry map (brighter = thicker); and (**d**) dilated skeleton (brown) with superimposed branching regions. Intersecting ones are united; (**e**): Volume rendering of the same sub-volume. Fiber-bundles (brown) with skeleton (green) and branching regions (orange); (**f**): Histogram of branching valences for the sub-sample a–e. Here, Y nodes (3 branches) clearly dominate.

**Figure 10 materials-14-01894-f010:**
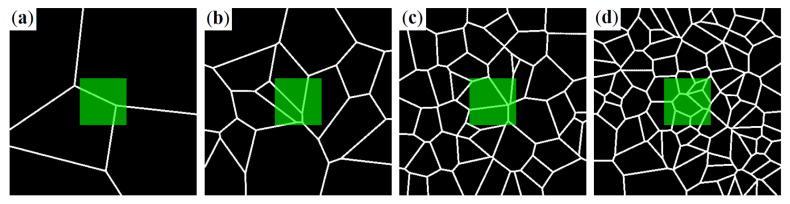
Realizations of 2D networks in images of 400 × 400 pixels. The expected number of generators or cells in this area λ is varied: (**a**) 10; (**b**) 20; (**c**) 40; and (**d**) 80. The green overlay indicates the 100 × 100 pixel sub-region finally used.

**Figure 11 materials-14-01894-f011:**
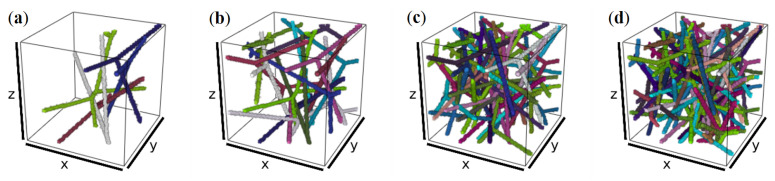
Rotated, superposed 2D network realizations forming a 3D network in images of $100^{3}$ voxels. $\lambda =10/400^{2}$. Varying ν yields varying solid fiber volume rates (VR): (**a**) ν=5, VR = 1%; (**b**) ν=15, VR = 3%; (**c**) ν=32, VR = 7%; (**d**) ν=45, VR = 10%.

**Figure 12 materials-14-01894-f012:**
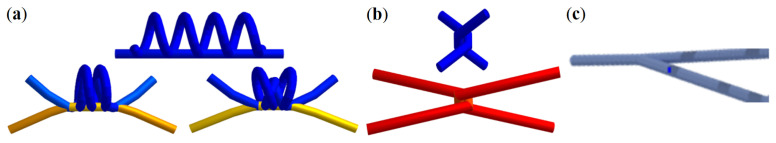
Representative winding/branching elements in leather meshes: (**a**) a coiled knot with (**left**) high and (**right**) low friction; (**b**) a simple contact interaction; and (**c**) typical Y-branching node.

**Figure 13 materials-14-01894-f013:**
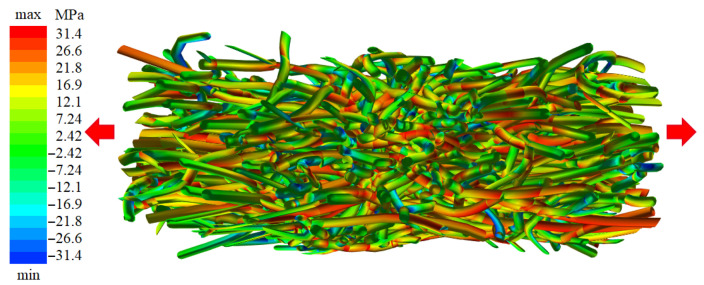
Reorganization of leather structure during horizontal tension test. Simulation was performed on a 5 mm cubic specimen with volume fraction of 63% (L3: zebu back). Colors indicate axial stress values from minimal (blue) to maximal (red).

**Figure 14 materials-14-01894-f014:**
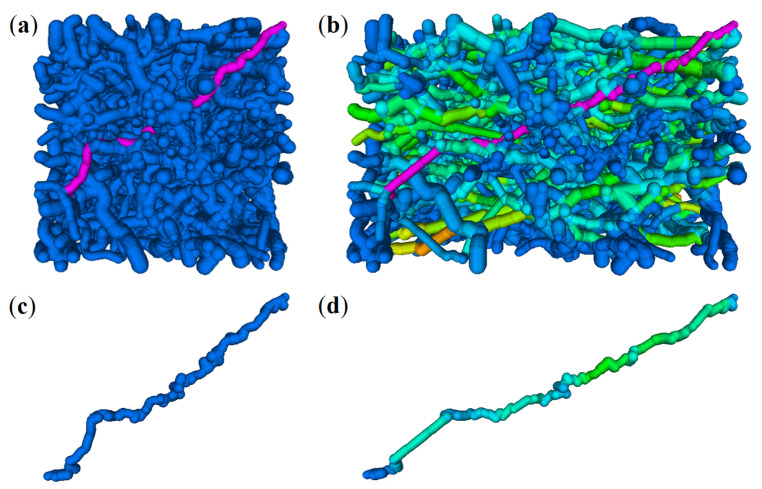
Visualization of a single fiber-bundle from L1, back with volume fraction 61%. (**a**,**c**): simulation start, (**b**,**d**): pulled state. (**a**,**b**): complete mesh with a marked thread; (**c**,**d**): the marked thread, isolated.

**Figure 15 materials-14-01894-f015:**
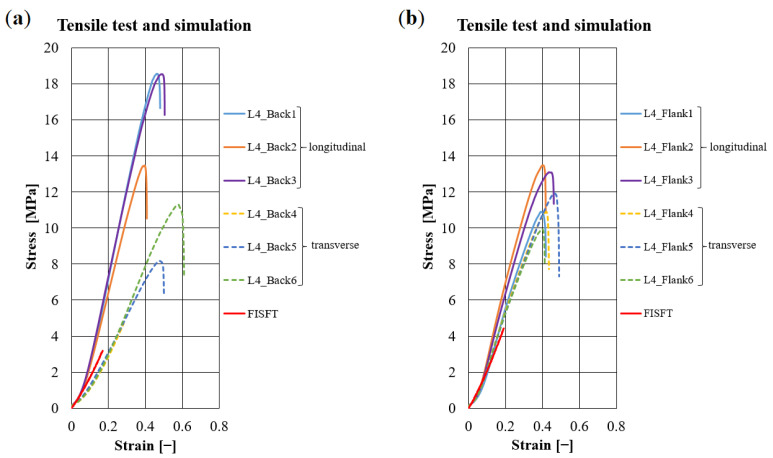
Comparison of tensile test results with the numerical simulations on model meshes fit to the synthetically tanned leather L4: (**a**) back and (**b**) flank.

**Figure 16 materials-14-01894-f016:**
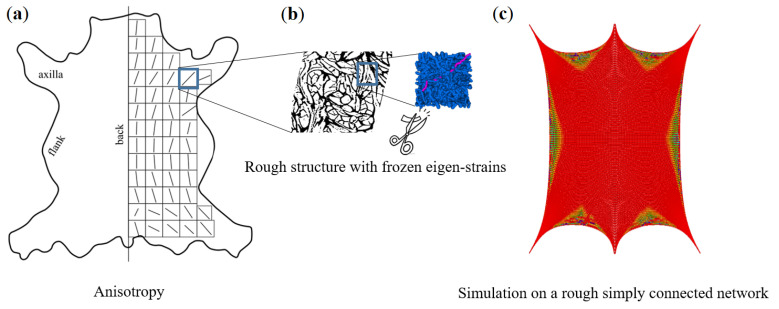
Hypothesis for a hide to be a connected network of locally differently prestressed fiber-bundles. (**a**) A schematic orientation of long fiber bundles under introduced pulling directions; (**b**) A selected long-fiber network turns to an isotropic meso-structure due to cutting during the preparation of CT-specimens; (**c**) Simulations of a frozen pre-stress in a fully connected network due to fixation at six directions in an animal (local fiber reorientation can be seen).

**Figure 17 materials-14-01894-f017:**
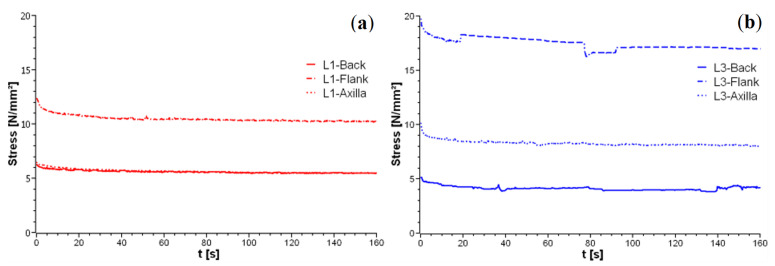
Relaxation curves of fiber-bundles for leather samples: (**a**) L1 and (**b**) L3 with 5%-elongation of initial length.

**Figure 18 materials-14-01894-f018:**
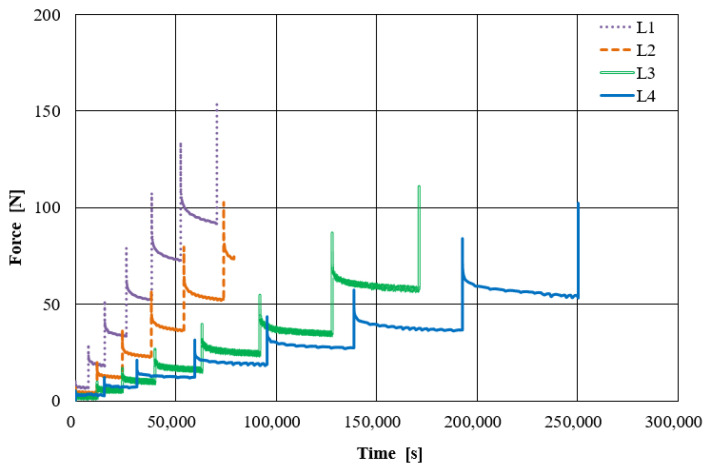
Exemplary incremental tensile test including determination of relaxation properties of leather samples.

**Figure 19 materials-14-01894-f019:**
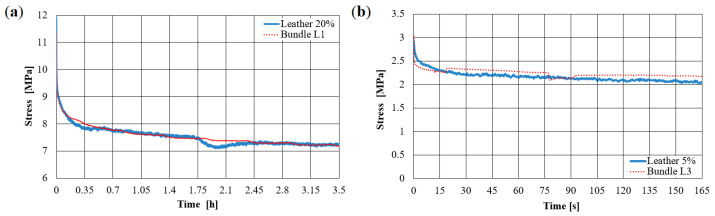
Comparison of relaxation curves calculated from fiber-bundle properties and from CT images (red curves) with relaxation tests for leather samples (blue curves): (**a**) Simmental breed L1 and (**b**) zebu L3.

**Figure 20 materials-14-01894-f020:**
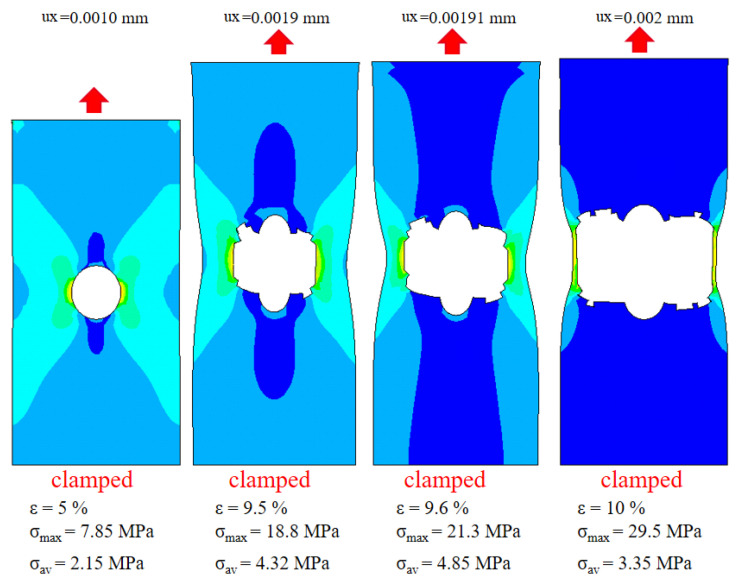
Simulated damage of a leather sample near a hole under vertical tension modelled using solid elements.

**Figure 21 materials-14-01894-f021:**
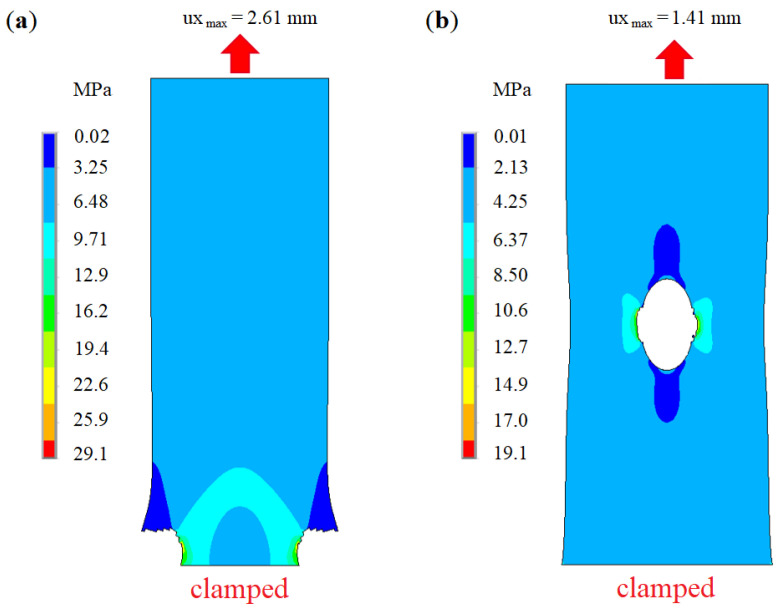
Von Mises stress distribution in a tensile FEM simulation of the damage of the leather stripes of Simmental breed (**a**) without and (**b**) with initial damage.

**Table 1 materials-14-01894-t001:** Summary of physical and mechanical measurements for leathers L1–L4.

Leather	Part	Thickness, mm	Mass per Area, g/${\mathrm{cm}}^{2}$	CT Based Volume Fraction, %	$F_{\max}$, N	*E*, MPa	σ, MPa	ϵ, %
L1	back	1.60	1095	61	178	53.0	10.8	26.3
	flank	1.56	893	52	102	24.1	5.82	29.4
	axilla	1.82	905	58	89	30.6	5.59	22.5
L2	back	1.66	1130	63	196	49.4	11.5	29.3
	flank	1.76	1103	62	174	38.5	9.83	31.5
	axilla	1.74	881	61	143	32.1	9.19	38.9
L3	back	1.63	1069	63	126	39.9	7.59	24.5
	flank	1.81	970	55	139	28.1	7.68	37.7
	axilla	1.86	978	54	123	29.6	7.51	34.6
L4	back	1.42	1004	66	185	34.3	13.3	49.1
	flank	1.35	945	68	159	33.1	11.7	41.8
	axilla	n.a.	n.a.	n.a.	n.a.	n.a.	n.a.	n.a.

n.a.—not available.

**Table 2 materials-14-01894-t002:** Experimental results for tension tests on fiber-bundles.

Leather	Localization	Thickness, mm	$F_{\max}$, N	σ, MPa	*E*, MPa	ϵ, %
L1	back	0.156	258	13.6	304	7.90
	flank	0.128	45.3	3.54	210	2.90
	axilla	0.147	3.5 *	0.21 *	4.96 *	6.26
L2	back	0.107	51.7	6.46	308	5.20
	flank	0.095	44.4	6.22	264	4.50
	axilla	0.039	33.5	27.7	517	2.38
L3	back	0.153	131	7.11	216	6.44
	flank	0.134	434	30.8	233	11.8
	axilla	0.120	342	30.3	235	3.34
L4	back	0.118	652	23.7	294	14.6
	flank	0.106	731	26.0	190	20.2
	axilla	0.116	323	12.8	130	21.4

* Outliers not used for the numerical simulations.

**Table 3 materials-14-01894-t003:** Tensile test data of leather samples.

			Back (Average Value)			Flank (Average Value)	
Leather	Size, mm	Thickness, mm	*E*, MPa	Strength *, MPa	Thickness, mm	*E*, MPa	Strength *, MPa
L1	10 × 50	1.64	53.0	12 (15)	1.69	24.1	5.5 * (8)
L2	10 × 50	1.71	49.4	9 * (13.2)	1.78	38.5	6 * (11.5)
L3	10 × 50	1.66	39.9	6 * (9)	1.82	28.1	4 * (9)
L4	10 × 50	1.38	34.3	10 ** (13.3)	1.36	33.1	9 ** (11.7)

*, **—stress at a strain of about 20%, 30%; ()—maximal stresses.

**Table 4 materials-14-01894-t004:** Simulation data for leathers.

Hole Diam., mm	Size *x, y, z*, mm	$E_{x}$/$E_{y}$($E_{z}$), MPa	G, MPa	μ, -	Elongation, mm	Aver. Stress, MPa	Max. Stress (Concentr.), MPa
Simmental breed (limit strength 10 MPa)				
3	20 × 10 × 1.77	55/45	20	0.3	1.41	6.5	19.1
-	20 × 10 × 1.77	55/45	20	0.3	2.61	4.25	29.1
Zebu (limit strength 6.5 MPa)				
3	20 × 10 × 1.77	45/30	15	0.3	1.15	3.5	15.7
-	20 × 10 × 1.77	45/30	15	0.3	1.51	2.2	9.89

## Data Availability

Data is contained within the article.

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
