# Peer review of "Simulation of Leather Visco-Elastic Behavior Based on Collagen Fiber-Bundle Properties and a Meso-Structure Network Model"

_materials, 2021, doi:10.3390/ma14081894_

Round 1

Reviewer 1 Report

The Article is devoted to computer modeling of strength and viscoelastic properties of natural leather. The subject of the Article is consistent with the topic of the Materials Journal and, given the numerous uses of leather products in various fields, it may be of interest to a wide audience of readers, especially to leather companies. The Authors' approach is based on modeling a network of connected collagen fiber bundles with contacts or branches. The calculations are carried out within the framework of the model of intertwined systems of extended edges of two-dimensional Poisson-Voronoi tessellations. To obtain the parameters necessary for the calculation, the Authors use the results they obtained using light and electron microscopic imaging, study of micro-mechanical properties of collagen-field bundles, and 3D microcomputer tomography of skin fragments. Using structural modeling, the Authors obtained good agreement between the experimental and calculated dependences of tensile tests, viscoelastic properties and relaxation tests of leather in different scales.

In conclusion, the Article is of interest for Materials, it is clearly written, properly presented, and can be published after minor revision. Some remarks are in pdf -file.

Reviewer 2 Report

Dear authors,   

People use products of the leather-processing industry on a daily basis. These include especially shoes, leather, and textile goods. The primary raw material for final products is hide from animals from slaughterhouses and hide from game–i.e. waste from the meat industry, which is processed in tanneries and turned into leather. Therefore, the tanning industry can be considered one of the first industries to use and recycle secondary raw materials. Although the tanning industry is environmentally important as a principal user of meat industry waste, the industry is perceived as a consumer of resources and a producer of pollutants. Processing one metric ton of raw hide generates 200 kg of final leather product (containing 3 kg of chromium), 250 kg of non-tanned solid waste, 200 kg of tanned waste (containing 3 kg of chromium), and 50,000 kg of wastewater (containing 5 kg of chromium) Thus, only 20% of the raw material is converted into leather. Besides, more and more people tend to use natural products avoiding chemical synthesis products, which leads to a controversial debate about the use of leathers.

            In general, the article is well written, the methods are explained correctly, but I recommend avoiding personal forms (personal writing, examples: we are ...; we investigate...; we conclude...), and also I recommend to avoid to use the structure "We present here for the first time ...", the article should be highlighted and not the people who write it. Every year, several hundred studies about leather are published with different topics (the sustainability of the leather, properties, and other adjacent topics); this shows that this article has value, but significant changes are needed before publication.

  • Lines 1-9 – Abstract "We present here for the first time a complete workflow for assessing leather quality by multi-scale simulation of elastic and visco-elastic properties." I recommend using the format recommended by the journal. "The abstract should be a single paragraph and should follow the style of structured abstracts, but without headings: 1) Background: Place the question addressed in a broad context and highlight the purpose of the study; 2) Methods: Describe briefly the main methods or treatments applied. Include any relevant preregistration numbers, and species and strains of any animals used. 3) Results: Summarize the article's main findings; and 4) Conclusion: Indicate the main conclusions or interpretations."
  • Line 34 – "…CT image data is 34 reported by [7] and [8]." Who is 7 and who is 8? "[7] compared the 3D..." Who is 7?
  • Line 37 – "Here, we therefore introduce a novel..." I recommend adding a comma after "therefore", and I strongly recommend avoiding a formulation like this form.
  • Lines 47, 50, 51, 52 – I recommend using the same format for the whole manuscript according to the Instructions for Authors.
  • It is also essential to be mention if the experiments are in triplicate (completed with the standard deviation).
  • Line 186 – I recommend modifying the sentence.
  • The manuscript format needs to be divided by The Materials and Methods ("described with sufficient details to allow others to replicate and build on the published results. Please note that the publication of your manuscript implicates that you must make all materials, data, computer code, and protocols associated with the publication available to readers". A part of the discussion is mandatory to be added. Authors should discuss the results and how they can be interpreted from previous studies and the working hypotheses. The findings and their implications should be discussed in the broadest context possible. Future research directions may also be highlighted. The discussion part is mandatory to be added. Authors should discuss the results and how they can be interpreted from previous studies and the working hypotheses. In addition, different articles from different authors need to be compared with the results obtained.
  • In final, I totally disagree with self-citations by authors, and I strongly recommend avoiding this practice (example: Julia Orlik – cited by 15 times from a total of 35 references/ Katja Schladitz – cited by 7 times)

Reviewer 3 Report

The study throws light on the visco-elastic behaviour of fibre bundles. Well planned models and procedures.

Author Response

We thank the Reviewer3 for his/her evaluation and ask to see the current version of our manuscript which contains new corrections in red color.

Reviewer 4 Report

The authors use advanced mathematic modeling to simulate elastic and viscoelastic properties of leather and thus assess its quality. The design and content of this study are described in details. However, the reviewer is confused with the organization of the manuscript, which causes further understanding of the study. 

  1. The authors should add a "Materials and Methods" section. The description in 2. and 3. related to how to conduct the material characterization should be separated out and listed in the specific section. For example, Line 69-74 and Line 89-93 should be described in a separate section. The other description in 2. and 3. should be compiled to a section of "results and discussion". Modeling may be combined with discussion or written in a separate section.
  2. Too many figures (22) are included in the manuscript. The authors should follow the criteria of the journal regarding the number limitation of figures and tables. In general, the maximum number of figures in a manuscript should not be over 10. The figures in the sections of 3~6 showing the simulating process rather than the results should be better removed from the manuscript, or the authors may generate a supplementary material including all the figures that are not necessary to be present in the manuscript. 
  3. The conclusion should not have any citation. 
  4.  The unit bars in Figure 2 are too small to identify.
  5.  The authors should include a simple introduction of viscoelastic properties of leather and current research background in this area. How is it relevant to the leather quality?

Round 2

Reviewer 2 Report

The authors have cautiously replied and fully revised the manuscript as the suggestions and comments of the reviewer. Therefore, this work should be recommended for publication in this journal.

Reviewer 4 Report

Accepted in present form.